# Antibiotic (Mis)Use in COVID-19 Patients before and after Admission to a Tertiary Hospital in Serbia

**DOI:** 10.3390/antibiotics11070847

**Published:** 2022-06-24

**Authors:** Aleksa Despotović, Aleksandra Barać, Teodora Cucanić, Ksenija Cucanić, Goran Stevanović

**Affiliations:** 1Faculty of Medicine, University of Belgrade, 11000 Belgrade, Serbia; aleksandrabarac85@gmail.com (A.B.); goran_drste@yahoo.com (G.S.); 2Department of Clinical Pharmacotherapy, Clinic for Infectious and Tropical Diseases, University Clinical Centre of Serbia, 11000 Belgrade, Serbia; teodorac995@gmail.com (T.C.); xeniac995@gmail.com (K.C.)

**Keywords:** COVID-19, Serbia, antibiotic use, cephalosporins, adults

## Abstract

Antimicrobial resistance (AMR) is a global concern, and antibiotic use has risen throughout the COVID-19 pandemic. Up to 75% of COVID-19 patients are treated with antibiotics despite little evidence for their use. A retrospective study from 6 March 2020 (the start of the pandemic in Serbia) to 31 December 2021 was conducted at the Clinic for Infectious and Tropical Diseases, University Clinical Centre of Serbia. In total, 523 patients with a microbiological diagnosis of COVID-19 were included. Patient data were analysed, including antibiotic use before and after admission. Pre-admission use of antibiotics for COVID-19 treatment was documented in more than half of patients (58.1%), of which a third (34.1%) used more than one antibiotic. Macrolides, cephalosporins, and fluoroquinolones were mainly used, most frequently among patients aged between 31–45 years (75.2%). Prior antibiotic use was associated with a longer duration of illness at admission (8.8 vs. 5.7, *p* < 0.001), oxygen therapy upon admission (27.6% vs. 16.0%, *p* = 0.002), and a lower vaccination rate (60.7% vs. 50.7%, *p* = 0.04). When hospitalised, 72.1% of patients received antibiotics, primarily cephalosporins (71.9%). Significant efforts are needed to reduce antibiotic use in the community and improve prescribing rates by healthcare professionals.

## 1. Introduction

Antimicrobial resistance (AMR) is responsible for five million deaths every year and its contribution to global mortality will only continue to grow in the coming decades [1]. The COVID-19 pandemic has facilitated the problem of AMR through significant increases in antibiotic prescribing. Up to 75% of patients with COVID-19 are treated with antibiotics, even though the rate of coinfections is low and no clinical benefit has been confirmed [2,3,4]. Most countries in the European Union (EU) have managed to reduce antibiotic prescribing in the years prior to the pandemic [5], but the opposite is true for Serbia [6]. Higher AMR rates are seen in Serbia compared to almost all EU countries in the latest European Centre for Disease Prevention and Control (ECDC) and the World Health Organisation (WHO) reports [7]. For example, Serbia is in the group of several countries with the highest prevalence rate (25–50%) of *Staphylococcus aureus* isolates resistant to methicillin (MRSA) and is one of a very few countries where carbapenem resistance in *Pseudomonas aeruginosa* and vancomycin resistance in *Enterococcus faecium* isolates are detected in ≥50% strains, while the majority of European countries report rates of <25% [7]. In Serbia, it is illegal to buy antibiotics without a prescription, but the sale of antibiotics without a prescription in private pharmacy chains, as well as self-medication, represent significant gaps in better antimicrobial stewardship [8], which is required to reduce the burden of AMR, both in the community and in the hospital setting.

Through our retrospective study covering over 500 patients, we sought to investigate antibiotic use among patients hospitalised with mild-to-moderate COVID-19, before and after admission. We further investigated antibiotic use among major age groups and analysed various clinical features with respect to prior antibiotic use.

## 2. Results

The clinical characteristics of 523 patients included in the study are shown in Table 1. The mean age was 56.7 years and 214 (40.9%) patients were female. A large proportion of patients (325, 62.1%) had at least one comorbidity, most frequent being hypertension (227, 43.4%), obesity (138, 26.4%), and diabetes (70, 13.4%). Coinfections were present in 22 (4.0%) patients, half of which were urinary tract infections (UTIs; 10, 45.4%). 132 patients (25.2%) were vaccinated. The majority of patients recovered (476, 91.0%).

More than half of admitted patients (304, 58.1%) used antibiotics prior to admission, with differences in use across major age groups (Figure 1). The lowest rates were seen in patients ≥66 years (48.5%), whereas the highest rate of prior antibiotic use was observed for the 31–45-year group (75.2%).

The most frequently used antibiotics used for COVID-19 treatment were macrolides (32.4%) cephalosporins (29.6%) and fluoroquinolones (28.2%). A third of patients (105, 34.5%) reported using more than one antibiotic. When analysing prior antimicrobial use across major age groups, three antibiotic groups emerged with a similar percentage of use (Figure 2). Cephalosporin use ranged from 27.4% in 46–65 years to 34.8% in 18–30 years, macrolide use ranged from 30.4% in 18–30 years to 33.3% in 31–45 years, whereas fluoroquinolones use ranged from 21.7% in 18–30 years to 30.6% in 66+ years.

Various patient characteristics at admission were analysed in relation to prior antibiotic use. Associations with a longer duration of illness at admission (8.8 ± 5.3 vs. 5.7 ± 3.7, *p* < 0.001), a more frequent use for oxygen therapy upon admission (27.6% vs. 16.0%, *p* = 0.002) and a lower vaccination rate (60.7% vs. 50.7%, *p* = 0.04) were found (Table 2). Additionally, male gender (62.7% vs. 51.4%, *p* = 0.009) and younger age (55.1 ± 15.8 vs. 59.0 ± 16.2, *p* = 0.006) were also associated with prior antibiotic use. Several symptoms on admission, including fever (95.7% vs. 79.0%, *p* < 0.001), cough (78.0% vs. 58.0%, *p* < 0.001), malaise/fatigue (72.0% vs. 62.6%, *p* = 0.019), diarrhoea (22.0% vs. 15.1%, *p* = 0.044) and nausea (21.0% vs. 12.3%, *p* = 0.009) were more frequently encountered in patients who used antibiotics before admission.

The number of patients who received antibiotics after admission was even higher (377, 72.2%), with slightly lower differences in prescribing across age groups (Figure 3). The lowest rate of prescribing was seen in the 18–30-year group (60.7%), with the highest rates among patients ≥ 66 years (76.0%) and for patients 46–65 years old (74.7%).

More than one antibiotic was used in 89 (23.6%) patients after hospitalisation, and the vast majority of patients received cephalosporins (276, 71.9%), of which ceftriaxone (231, 83.7%) was by far the most frequently prescribed (Figure 4). Cefuroxime and cefepime were other notable cephalosporins, though used in much lower rates (in 23.1% and 15.5% of patients, respectively). Fluoroquinolone (49, 9.4%) and carbapenem (24, 4.7%) prescribing was significantly lower compared to cephalosporins. When observing post-admission prescribing across age groups, cephalosporin use was highest in the 18–30 year group (428, 81.8%).

## 3. Discussion

This is the first study from Serbia looking at antibiotic use in COVID-19 patients, revealing a very high prescribing rate. Despite the fact that only a small number of patients have coinfections (4.0% in our case), our results support numerous studies indicating disproportionately high rates of antibiotic prescribing without a clear indication [9,10,11,12]. Antibiotic use in Serbia is plagued by the poor health education of the general public which leads to frequent self-medication and overall misuse [8,13,14,15]. It is not surprising that overall antibiotic use has increased in the past decade in Serbia compared to virtually all countries in the European Union (EU) [6], with significant efforts needed on a national level to curb the development of AMR.

In our study, pre-admission antibiotic use was associated with a longer duration of illness before admission, a lower vaccination rate, and a more frequent need for oxygen therapy. Such findings hint at either self-treatment or a lack of awareness of the reasons for antibiotic usage, reinforcing the case for health education and more rigorous antibiotic access in the outpatient context. In the outpatient context, we discovered that the age group 31–45 received more antibiotics than the age group > 66. This might be explained by a lack of medical education, the minimization of subjective symptoms, the failure to report to the doctor, and, as a result, the freedom to treat oneself. A different symptom profile, including more frequent nausea and diarrhoea in patients who used antibiotics prior to admission, requires further analysis. This might be explained by the fact that persons with more severe symptoms, such as nausea and diarrhoea, sought out medications, including antibiotics, in the hope that they would alleviate their symptoms.

Regardless of the fact that antibiotics have no clinical benefit for COVID-19 patients and their use should be avoided unless indicated [3,16,17,18], in-patient prescribing rates reach as high as 75% in numerous countries [2], as was the case in our study. Serbian guidelines for COVID-19, from its initial versions, have clearly advised not to use antibiotic treatment unless there is a clear indication (Appendix A), but we still found very high rates of prescribing, particularly of cephalosporins. Although inappropriate overprescribing could be the primary reason for such high rates, it is still the primary drug of choice for empirical treatment of community acquired pneumonia, predominantly caused by *Streptococcus pneumoniae,* while atypical pneumonia is quite rarely encountered in our practice. Furthermore, the absence of an accompanying drug such as a macrolide can be explained by its overall avoidance in clinical practice due to high resistance rates, seen both in Serbia and in neighbouring countries, [6,7,19,20]. De-escalation of therapy after excluding pneumococcal pneumonia through microbiological testing, however, is a significant challenge at our hospital, as gold-standard bronchoalveolar lavage (BAL) is not available. Thus, improving microbiological capabilities would certainly optimise antibiotic prescribing, particularly early de-escalation. Nevertheless, further efforts are needed to prevent antibiotic misuse in the hospital setting and studies in Canada and the US have shown successful reductions in antibiotic prescribing [21,22]. In our country, it seems evident that recommendations for antibiotic prescribing and use are not being followed. More rigorous mechanisms of control and surveillance are needed, coupled with education of the general public, but also healthcare professionals.

Our study had a few important limitations. First, the single-centre study prevents the ability to generalize results on a population level. Second, the retrospective design of the study limited us in analysing pre-admission use of antibiotics, including duration of use, the mode of acquiring antibiotics and the indications (if present) for use. The same limitation applies for post-admission use, where a better understanding of in-hospital prescribing was prevented by the study design.

## 4. Materials and Methods

This retrospective study was conducted at the Department of Clinical pharmacotherapy of the Clinic for Infectious and Tropical Diseases, University Clinical Centre of Serbia. Throughout the COVID-19 pandemic, the Department of Clinical pharmacotherapy was responsible for treating mild-to-moderate patients suffering from COVID-19. Doctors in Serbia are obliged to follow the National Guidelines for Antimicrobial therapy [23], which was used as the main guideline in this study, in addition to the Serbian National COVID-19 Treatment Guidelines that were introduced since the beginning of the pandemic (Appendix A). From 6 March 2020 (the beginning of the COVID-19 pandemic in Serbia) to 31 December 2021, a total of 523 patients with a microbiological diagnosis of COVID-19 who were hospitalised and treated at our department were included in the study. Patients with incomplete medical records and those in whom a microbiological diagnosis was not made were excluded from the study.

Patient data were collected from the hospital’s electronic medical records system and the following variables were extracted: age, gender, vaccination status, presence of comorbidities and coinfections, length of stay (LOS), and outcome. Since the start of widespread vaccination in Serbia on 19 January 2021, three types of vaccines have become available, with the possibility of individual choice: messenger RNA (mRNA), viral vector vaccines, and inactivated vaccines. Information about antibiotic use, prior to and after hospitalisation, was collected as well, including the number of antibiotics used upon admission.

Mean and standard deviation was used to describe data with normal distribution, whereas median with minimum and maximum values was used for data that were not normally distributed. For categorical variables, numbers and percentages were used. The Chi-square test and the independent *t*-test were used to compare variables among patients who used antibiotics prior to admission. The level of significance *p* was established at <0.05.

## 5. Conclusions

More than half of patients used antibiotics prior to admission, while 72.2% of patients were prescribed antibiotics upon admission despite a very low number of coinfections and no evidence for clinical benefit. As the problem of AMR continues to grow, significant efforts are needed to limit antibiotic use in the community and educate the general public about the proper use of antibiotics. The same applies to the healthcare setting, where better strategies are needed to ensure adherence to guidelines for rational antibiotic prescribing.

## Figures and Tables

**Figure 1 antibiotics-11-00847-f001:**
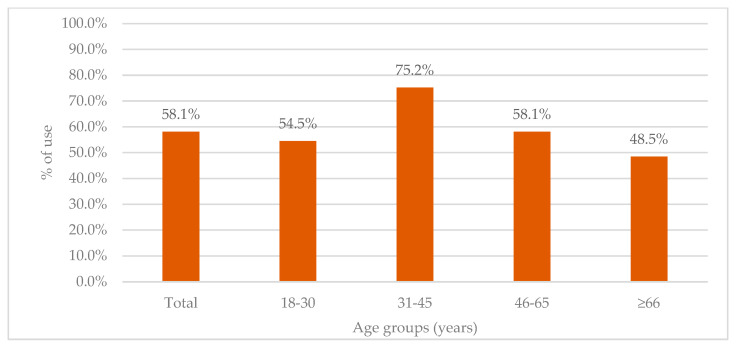
Pre-admission antibiotic use stratified across age groups.

**Figure 2 antibiotics-11-00847-f002:**
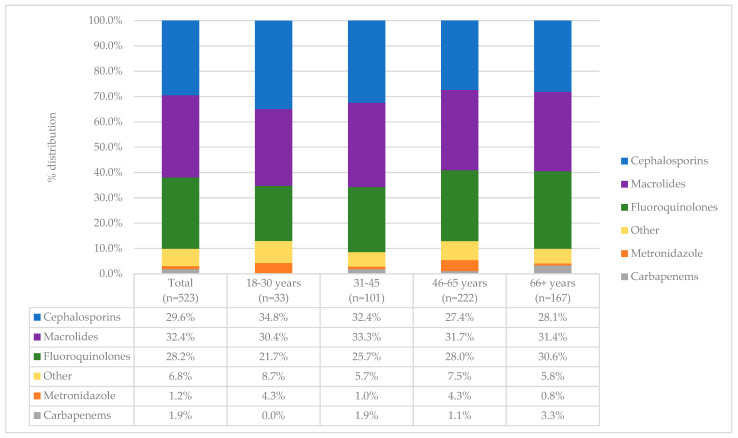
Pre-admission profile of antibiotics used across major age groups.

**Figure 3 antibiotics-11-00847-f003:**
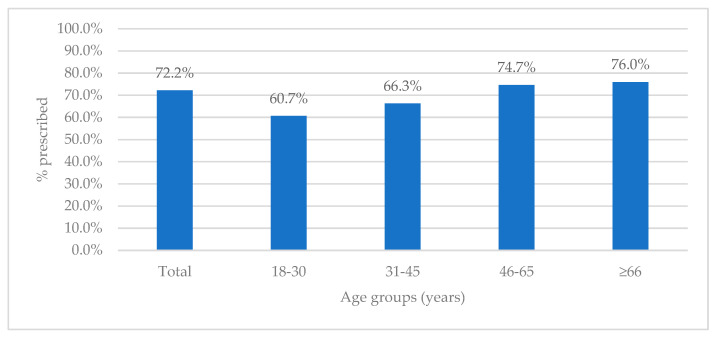
Antibiotic prescribing upon admission across age groups.

**Figure 4 antibiotics-11-00847-f004:**
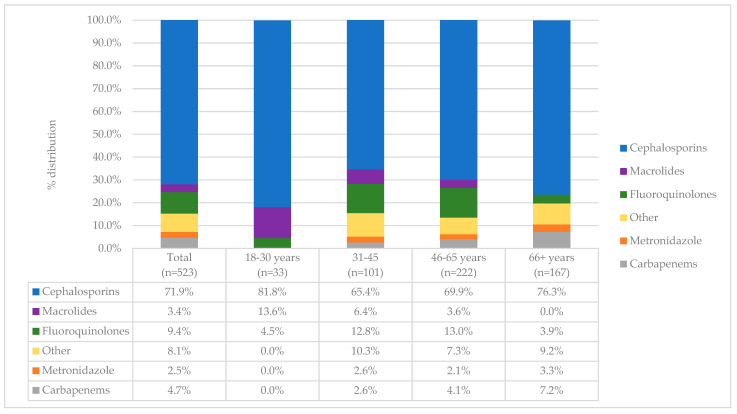
Post-admission profile of antibiotics used across major age groups.

**Table 1 antibiotics-11-00847-t001:** Clinical characteristics of study patients with COVID-19.

Patient Characteristics (n = 523)	N (%)	Patient Characteristics (n = 523)	N (%)
Age (mean ± SD)	56.7 ± 16.0	Coinfections	22 (4.0)
Gender (female)	214 (40.9)	Bacterial	16 (72.7)
Vaccinated	132 (25.2)	UTI	10 (45.4)
LOS (median, IQR)	9 (5)	SSI	3 (13.6)
		*Clostridium difficile*	2 (9.0)
Comorbidities	325 (62.1)	Syphilis	1 (4.5)
Hypertension	227 (43.4)	Viral/Fungal	6 (27.2)
Obesity	138 (26.4)	Primary EBV infection	1 (4.5)
Diabetes	70 (13.4)	Hepatitis B	1(4.5)
Atrial fibrillation	40 (7.6)	CMV	1 (4.5)
Coronary heart disease	39 (7.5)	Herpes zoster	1(4.5)
Solid Tumour	42 (8.0)	Candidiasis	1 (4.5)
COPD	11 (26.2)	Pulmonary aspergillosis	1(4.5)
Connective Tissue Disease	31 (5.9)		
Neurological disease	30 (5.7)	Outcome	
Leukaemia/Lymphoma	23 (4.4)	Recovered	476 (91.0)
Cardiomyopathy	20 (3.8)	Transferred to ICU	20 (3.8)
Paralysis	19 (3.6)	Transferred to another hospital	19 (3.6)
Hashimoto thyroiditis	15 (2.9)	Died	3 (0.6)
Liver disease	10 (1.9)	Discharged at personal request	5 (1.0)
Chronic kidney disease	10 (1.9)		
Congestive heart failure	9 (1.7)		
Dementia	8 (1.5)		
HIV	6 (1.1)		
Peptic ulcer disease	4 (0.8)		

SD: standard deviation. LOS: length of stay; IQR: interquartile range; COPD: Chronic obstructive pulmonary disease; HIV: human immunodeficiency virus; UTI: urinary tract infection; SSI: skin and soft tissue infection; EBV: Epstein-Barr virus; CMV: cytomegalovirus; ICU: intensive care unit.

**Table 2 antibiotics-11-00847-t002:** Patient characteristics in relation to antibiotic use prior to admission.

Variables	Pre-Admission Antibiotic Use		
	Yes (n = 304)	No (n = 219)	*p*
Gender (male)	194 (62.7)	110 (51.4)	0.009
Comorbidities	180 (59.2)	145 (66.2)	0.103
Vaccinated	68 (50.7)	153 (60.7)	0.045
Age	55.1 ± 15.8	59.0 ± 16.2	0.006
Day of illness at admission (median, IQR)	8.0 (4)	5.0 (4)	<0.001
O_2_ therapy upon admission	84 (27.6)	35 (16.0)	0.002
Symptoms			
Fever	291 (95.7)	173 (79.0)	<0.001
Cough	237 (78.0)	127 (58.0)	<0.001
Malaise/fatigue	219 (72.0)	137 (62.6)	0.019
Dyspnoea	79 (26.0)	43 (19.6)	0.088
Myalgia	73 (24.0)	41 (18.7)	0.145
Diarrhoea	67 (22.0)	33 (15.1)	0.044
Nausea	64 (21.0)	27 (12.3)	0.009
Vomiting	22 (7.2)	12 (5.5)	0.418

IQR—interquartile range.

## Data Availability

Data supporting the results of this study are not publicly available but can be made available on request of the corresponding author.

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
