# Peer review of "Antibiotic (Mis)Use in COVID-19 Patients before and after Admission to a Tertiary Hospital in Serbia"

_antibiotics, 2022, doi:10.3390/antibiotics11070847_

Round 1
Reviewer 1 Report
1. The introduction could be expanded by giving specific examples of antimicrobial resistance rates in the EU available in the ECDC in different countries, as well as in Serbia, just to let the reader understand the magnitude of the AMR problem
2. Table 1: Please define what is shown in parentheses and with ±. For example Age, mean ±SD. Length of stay, median (IQR) [or (range)].
3. Table 1: Please define COPD in the footnote
4. Table 2: The way the day of illness at admission is presented implies that mean ±SD is shown and a t-test was performed. Did the data pass a normality test in that case? Otherwise, the median (IQR) should be shown and a Mann-Whitney test should be performed instead
5. Line 97: Here, a comparison of AMR in Serbia with the rest of the EU could be performed, as also stated in the first comment, with data from the ECDC
6. Something that was a little bit unexpected in the results section was the finding that antimicrobial use was more frequent in the 31-45 years old group (before admission). One would expect antimicrobial use to be more frequent in older patients. Any idea why was that noted?
7. A limitations subsection is needed at the end of the discussion section. For example, this is a single-center study
8. Line 123: The study does not completely clarify if all included patients were hospitalized. Please state if this is the case
9. Acknowledgements: It is empty, thus, please state ‘none’, if that is the case
10. References: Most references have double numbering. Please correct that
Author Response
Dear Reviewer,
Many thanks for taking the time to look at our manuscript and provide feedback. The responses to your comments are as follows:
1. The introduction could be expanded by giving specific examples of antimicrobial resistance rates in the EU available in the ECDC in different countries, as well as in Serbia, just to let the reader understand the magnitude of the AMR problem
We agree that this kind of information would be beneficial - Lines 32-38 now include specific examples that point to higher AMR in Serbia compared to EU countries, with an additional reference added reflecting the latest data. We hope that this addition is satisfactory, but if not, do let us know. Lines 39-42 give additional context to the situation.
2. Table 1: Please define what is shown in parentheses and with ±. For example Age, mean ±SD. Length of stay, median (IQR) [or (range)].
Thank you for pointing this out - age has been supplemented with "(mean ± SD)" , whereas Length of Stay has been supplemented with "(median, IQR)", and both LOS and IQR have been added to the Table 1 footnote.
3. Table 1: Please define COPD in the footnote
COPD has been added to the Table 1 footnote, we appreciate the comment.
4. Table 2: The way the day of illness at admission is presented implies that mean ±SD is shown and a t-test was performed. Did the data pass a normality test in that case? Otherwise, the median (IQR) should be shown and a Mann-Whitney test should be performed instead
Thank you for pointing out this valid comment - indeed, upon review, "day of illness at admission" fulfils only some, but not all criteria for normal distribution - for this reason, the Mann-Whitney test was performed, resulting in the same p-value of < 0.001. The change in its presentation (mean, SD > median, IQR) was made accordingly.
5. Line 97: Here, a comparison of AMR in Serbia with the rest of the EU could be performed, as also stated in the first comment, with data from the ECDC
Thank you for this suggestion - we have provided a more nuanced comparison of AMR in Serbia compared to other EU countries in the introduction, as you kindly pointed out. However, if you think additional comparisons are necessary in the discussion as well, we are happy to make additional changes to this part of the manuscript.
6. Something that was a little bit unexpected in the results section was the finding that antimicrobial use was more frequent in the 31-45 years old group (before admission). One would expect antimicrobial use to be more frequent in older patients. Any idea why was that noted?
Though surprising for us as well, we identified possible reasons for such findings. In the discussion, lines 118-121 now address this finding.
7. A limitations subsection is needed at the end of the discussion section. For example, this is a single-center study.
This is a rather poor oversight on our part, and we thank you for pointing this out. Lines 147-152 are now containing the "limitations" subsection.
8. Line 123: The study does not completely clarify if all included patients were hospitalized. Please state if this is the case
Yes, all patients were hospitalised, and the change is now made in Line 162.
9. Acknowledgements: It is empty, thus, please state ‘none’, if that is the case
Thank you for this comment - this section has now been populated (lines 207-209).
10. References: Most references have double numbering. Please correct that
Thank you for this information - references are now corrected and no double numbering is present.
Once again, we thank you for taking the time to review our manuscript. We hope our responses are satisfactory.
Sincerely, on behalf of the study team,
Aleksa Despotović

Reviewer 2 Report
The article describes the use of antibiotics in patient with mild-to-moderate COVID-19 before and after the admission to a hospital in Serbia.
Several major limitations can be mentioned both in the methodology and the redaction of the article:
Abstract and Introduction
- L. 8 "dramatically risen" and l. 27: "greatly facilitated the problem of AMR":
I would be less "dramatic" when describing AMR and antibiotic consumption in the abstract and introduction. The antibiotic consumption seems to have decreased in EU countries as you say and the lockdown measures (stay at home), increase of hand hygiene and travel restriction rather lead to a decrease or stability of AMR.
Do you have data from Serbia which shows an increase of AMR?
- L.14: history of prior antibiotic use: be more precise by using the terms (prior antibiotic use in case of COVID-19, antibiotic use for treating other infections is not included)
- L.35: I would suggest to add "investigate antibiotic use among patients hospitalized with mild-to-moderate COVID-19"
Method
- Globally this section insufficiently describes your methodology. Much more details should be add in regard of the introduction of the guidelines f.e.
- Setting: describe the hospital in which the study was conducted.
- Did you assess the appropriateness ? proportion of appropriate prescriptions ?
Result
- Did you observe a difference of use in the beginning of the pandemic and later ? Is the inappropriateness was particularly seen at the beginning of the pandemic ? did you observe an improvement after the introduction of guidelines ?
- L. 55: "most frequently used antibiotics…": please add for treating COVID-19.
Discussion
- Have you an explanation why the age group 31-45 received more antibiotics than age group >66 in the outpatient setting ?
- What are the lessons learned ? How can your results be extrapolated to other hospitals ?
- What are the limitations of the study ?
Author Response
Dear Reviewer,
We thank you for taking the time to review our manuscript and for the provided feedback.
We are providing answers to your comments as follows, with your initial comments:
Introduction
1. 8 "dramatically risen" and l. 27: "greatly facilitated the problem of AMR":
I would be less "dramatic" when describing AMR and antibiotic consumption in the abstract and introduction. The antibiotic consumption seems to have decreased in EU countries as you say and the lockdown measures (stay at home), increase of hand hygiene and travel restriction rather lead to a decrease or stability of AMR.
Do you have data from Serbia which shows an increase of AMR?
We acknowledge your comment related to the dramatisation of AMR, and have respectively deleted "dramatically" and "greatly" from lines 8 and 27. Antibiotic prescribing has indeed been improving in the EU over the past years (as we mention in line 30-31) and we did cite studies from Canada/US showing reductions in overall antibiotic use during the pandemic - but based on large-scale evidence that many patients treated for COVID-19 received antibiotics without an appropricate indication, we used that evidence as the basis for the argument. If you think we should make additional corrections in the introduction, we will do so.
In response to your second comment - lines 32-38 now compare AMR in Serbia compared to most EU countries and provide specific examples - the latest report from eCDC and WHO (cited under #7) clearly show a much higher prevalence of resistant isolates in our country. Some additional context follows, in lines 38-42.
2. L.14: history of prior antibiotic use: be more precise by using the terms (prior antibiotic use in case of COVID-19, antibiotic use for treating other infections is not included)
Thank you for pointing this out - line 14 has now been appropriately modified to be more specific about antibiotic use.
"Pre-admission use of antibiotics for COVID-19 treatment was documented..."
3. L.35: I would suggest to add "investigate antibiotic use among patients hospitalized with mild-to-moderate COVID-19"
Thank you for this piece of feedback - indeed, to be more specific about the patient population and the severity of COVID-19, we have added the additional description, now line 43-44.
Method
4. Globally this section insufficiently describes your methodology. Much more details should be add in regard of the introduction of the guidelines f.e.
Additional clarification regarding the guidelines used for overall antimicrobial use is now described in lines 158-161.
5. Setting: describe the hospital in which the study was conducted.
The description of the hospital can be found on lines 121 - 127. If this description is not satisfactory, kindly let us know what are additional pieces of information we need to provide.
6. Did you assess the appropriateness ? proportion of appropriate prescriptions ?
Unfortunately, the retrospective study design did not allow us to further investigate the appropriateness of prescribing rates, both pre-admission and post-admission. Apart from the information that only 4% of patients had coinfections (which does point, at least partially, to insufficient indications for use in a large proportion of patients), there was not much we could obtain for further analysis. However, your feedback is still relevant, and in conjunction with your last comment, this has been added to the last paragraph of the discussion, where the limitations of this study were described - lines 147-152.
Results
7. Did you observe a difference of use in the beginning of the pandemic and later? Is the inappropriateness was particularly seen at the beginning of the pandemic? did you observe an improvement after the introduction of guidelines ?
To our knowledge, this would be the first study from our country looking at antibiotic use in COVID-19 patients - we could potentially look into antibiotic use over time in our sample, but our sample would be too small to make any meaningful conclusions. On the basis of these findings, our next step would be to make a multicentric study with other departments at our hospitals (and other hospitals as well involved in treating COVID-19 patients) to further analyse antibiotic prescribing.
With respect to your comment about the introduction of guidelines, the very first version of national COVID-19 guidelines strongly advised against use of antibiotics without a clear indication. Therefore, any comparison in relation to guideline introduction wouldn't yield any meaningful results.
8. L. 55: "most frequently used antibiotics…": please add for treating COVID-19.
Thank you for this comment, we have made the suggested correction.
Discussion
9. Have you an explanation why the age group 31-45 received more antibiotics than age group >66 in the outpatient setting ?
Thank you for this comment - we have added our response to this finding in lines 118-121.
10. What are the lessons learned ? How can your results be extrapolated to other hospitals?
We expanded our discussion in lines 143-146 to address this particular question - indeed, you were right in pointing out that we were lacking wider implications of our findings and potential benefit for other hospitals.
11. What are the limitations of the study?
Thank you for pointing out the absence of the section related to limitations in this manuscript. Lines 147-152 now contain information about the limitations of our study.
Once again, we thank you for taking the time to review our manuscript. We hope our responses are satisfactory.
Sincerely, on behalf of the study team,
Aleksa Despotović

Reviewer 3 Report
Despotovic et al. present a brief analysis of antibiotic use in Serbian patients that were admitted to the hospital with COVID-19 infections. Based on the study Introduction and Discussion, it seems as though the nation of Serbia will benefit from antimicrobial stewardship initiatives that stratify multiple tiers of society. The results of the current study may potentially be used to support future stewardship efforts in the country; however, I believe additional context about the study and current practice trends in Serbia may increase the impact of the manuscript.
· I recommend adding “Serbia” to the title of the manuscript “…to a Tertiary Hospital in Serbia” to clarify the focus of the manuscript.
· Lines 93 – 94 – can the authors comment on how the general public self-medicate for infections in Serbia? Is a prescription from a healthcare provider needed for antibacterials? In the Introduction it will be helpful to explain the most common sources of outpatient antibacterial use (ambulatory clinic versus prescription from a family medicine physician versus private pharmacy without a prescription, etc.) and maybe one sentence on cultural trends in antimicrobial use.
· Can the authors please talk a little about the clinical pathways or practice habits at the institution where the study occurred? Is there a local protocol that provides instructions for physicians to help determine when antibacterials are required empirically for a patient that has COVID-19? Also, are there suggestions for how to determine if a respiratory infection is bacterial or viral through ordering sputum cultures, urinary antigen tests, procalcitonin, etc?
· Do the authors know how long each patient was using outpatient antibacterials before being admitted to the hospital, and if so, can the average duration of outpatient therapy prior to admission please be reported as well?
· Presumably a lot of the antibacterial overuse in the hospital stemmed from antibacterials that were being used to cover for community-acquired bacterial pneumonia. In the 2019 IDSA guidelines on CAP, hospitalized patients should receive a regimen that covers for certain aerobic pathogens and also atypical pathogens such as Mycoplasma. In Figure 4, I am surprised that such a high percentage of patients received a cephalosporin without also receiving a second drug for atypical coverage such as a macrolide or doxycycline. Is there an explanation for why cephalosporins were used so frequently for hospitalized patients without an accompanying macrolide or doxycycline? Were the physicians only concerned about CAP from a pathogen like Streptococcus pneumoniae but felt confident that an atypical bacterial pneumonia could be distinguished from COVID-19?
· Cephalosporin use is such a large percentage of overall antibacterial use for hospitalized patients that it may be helpful to list in the text of the manuscript which cephalosporins specifically were used the most.
· Table 2 – have the authors considered a multivariate analysis to see which variables were independently associated with antibiotic use prior to admission?
· Which types of vaccines were most widely available in Serbia during the study window? It will be helpful to know if the vaccinated patients received mRNA vaccines versus another platform for example.
· Lines 102 – 106 – isn’t a likely explanation for the higher rates of gastrointestinal symptoms in patients that used antibacterials prior to admission that the antibacterials themselves were the cause of some of the nausea and vomiting?
· Line 58 – It seems like the paragraph actually corresponds to Figure 2, not Figure 1.
· The length of stay and outcome are both listed in Table 1 in an aggregated table of clinical characteristics of all the patients, but it would be interesting to compare the length of stay, outcome, frequency of adverse events (such as allergic reactions), and other variables between patients that did receive antibacterials during hospitalization and those that did not receive antibacterials. If the length of stay and outcomes are similar between the two groups, the authors can then demonstrate that the use of antibacterials was not associated with a benefit to the patient and may actually be harmful (if adverse effects of antibacterials are accessible in the database).
Author Response
Dear Reviewer,
We thank you for taking the time to review our manuscript and for the provided feedback.
We are providing answers to your comments as follows, with your initial comments in bold:
1. I recommend adding “Serbia” to the title of the manuscript “…to a Tertiary Hospital in Serbia” to clarify the focus of the manuscript.
"Serbia" has been added to the title of the manuscript
2. Lines 93 – 94 – can the authors comment on how the general public self-medicate for infections in Serbia? Is a prescription from a healthcare provider needed for antibacterials? In the Introduction it will be helpful to explain the most common sources of outpatient antibacterial use (ambulatory clinic versus prescription from a family medicine physician versus private pharmacy without a prescription, etc.) and maybe one sentence on cultural trends in antimicrobial use
Thank you for this comment - indeed, information surrounding the context of high AMR in Serbia is lacking. We have expanded the Introduction to provide specific examples of high AMR rates compared to EU countries, but more information on the reasons for misuse as well in lines 32-42. We hope this explanation is satisfactory.
3. Can the authors please talk a little about the clinical pathways or practice habits at the institution where the study occurred? Is there a local protocol that provides instructions for physicians to help determine when antibacterials are required empirically for a patient that has COVID-19? Also, are there suggestions for how to determine if a respiratory infection is bacterial or viral through ordering sputum cultures, urinary antigen tests, procalcitonin, etc?
Thank you for pointing out an important aspect of antibiotic prescribing that was not entirely taken into consideration in our submitted manuscript. We expanded our manuscript at multiple sites to respond to this question in full: description of the protocol used (lines 158-161), and the description of current (and insufficient) diagnostic practices (lines 137-41). These questions were partly answered along with your subsequent comment regarding the guidelines for CAP treatment as well (see below), so we hope we were ultimately able to address this particular comment in full.
4. Do the authors know how long each patient was using outpatient antibacterials before being admitted to the hospital, and if so, can the average duration of outpatient therapy prior to admission please be reported as well?
We agree that this would be of tremendous value for the quality of the manuscript, but unfortunately, the retrospective study design limited us in analysing pre-admission antibiotic use to that extent, including information regarding the duration of use, but also information related to the indication of use and mode of acquisition (prescription vs. self-medication, etc.). This limitation is now located in the manuscript, lines 147-152.
5. Presumably a lot of the antibacterial overuse in the hospital stemmed from antibacterials that were being used to cover for community-acquired bacterial pneumonia. In the 2019 IDSA guidelines on CAP, hospitalized patients should receive a regimen that covers for certain aerobic pathogens and also atypical pathogens such as Mycoplasma. In Figure 4, I am surprised that such a high percentage of patients received a cephalosporin without also receiving a second drug for atypical coverage such as a macrolide or doxycycline. Is there an explanation for why cephalosporins were used so frequently for hospitalized patients without an accompanying macrolide or doxycycline? Were the physicians only concerned about CAP from a pathogen like Streptococcus pneumoniae but felt confident that an atypical bacterial pneumonia could be distinguished from COVID-19?
This is a really interesting piece of feedback, and thank you for this comment. As the story of antibiotic prescribing gets more complicated under the circumstances of COVID-19 treatment, it was difficult for us to retrospectively analyse the "appropriateness" of antibiotic choice, especially in our context where we observed prescribing of predominantly cephalosporins. However, on the basis of your comment and our local practices, we expanded the discussion by providing a more detailed explanation related to cephalosporin use, including limited ability to perform proper de-escalation therapy (lines 131-141). We hope the discussion, in its amended form, is able to address your comment.
6. Cephalosporin use is such a large percentage of overall antibacterial use for hospitalized patients that it may be helpful to list in the text of the manuscript which cephalosporins specifically were used the most.
Thank you for this suggestion - we have provided additional information in lines 95-98.
7. Table 2 – have the authors considered a multivariate analysis to see which variables were independently associated with antibiotic use prior to admission?
Yes, this has been a consideration, but we decided against conducting such an analysis. The main reason is that, apart from qualitative knowledge about prior antibiotic use, we didn't have any additional data points such as duration of use and mode of acquisition. Therefore, we decided to stick to a more "basic" presentation of information, but your comment did get brought up in recent discussions about a broader study that would obtain results necessary for such an analysis.
8. Which types of vaccines were most widely available in Serbia during the study window? It will be helpful to know if the vaccinated patients received mRNA vaccines versus another platform for example.
Thank you for this comment and we agree that it would be beneficial to add more information surrounding vaccination availability. We have provided additional information regarding vaccination in the methods section (lines 169-172) - if, however, you deem that there is a more appropriate place for it, we are happy to make subsequent changes.
9. Lines 102 – 106 – isn’t a likely explanation for the higher rates of gastrointestinal symptoms in patients that used antibacterials prior to admission that the antibacterials themselves were the cause of some of the nausea and vomiting?
We have considered this explanation and consider it to be likely as well. But, since we do not possess information about the date/duration of antibiotic use, nor do we know when did gastrointestinal symptoms start, we are unable to make a claim about the nature of the connection between antibiotic use and gastrointestinal symptoms. Furthermore, this connection becomes more complicated when we consider very high rates of post-admission antibiotic use, meaning we would also have to analyse the persistence of GI symptoms. We hope this explanation is satisfactory, but we are welcome to make any modifications to the manuscript if you deem it necessary.
10. Line 58 – It seems like the paragraph actually corresponds to Figure 2, not Figure 1.
This is, indeed, correct, and thank you for this comment. Appropriate correction has been made.
11. The length of stay and outcome are both listed in Table 1 in an aggregated table of clinical characteristics of all the patients, but it would be interesting to compare the length of stay, outcome, frequency of adverse events (such as allergic reactions), and other variables between patients that did receive antibacterials during hospitalization and those that did not receive antibacterials. If the length of stay and outcomes are similar between the two groups, the authors can then demonstrate that the use of antibacterials was not associated with a benefit to the patient and may actually be harmful (if adverse effects of antibacterials are accessible in the database).
This is a valid point, and we thank you for bringing this up - When we initially considered publishing these results, and similarly to your earlier comments about conducting a regression analysis on pre-admission use, we also discussed further analysis of post-admission use and outcomes. We concluded that, for the analysis to be valid, we would have to adjust in-hospital use by a large number of factors, the most important being other drugs that patients received throughout their hospitalisation. As many patients received corticosteroids and/or antivirals and/or symptomatic therapy (at different times throughout the hospitalisation, too), this analysis would be challenging to perform. In addition to the treatment (and a number of demographic factors, of course), the severity of illness at admission would have to be taken into account, for which we have some, but not all the necessary information. For these reasons, we decided to limit this investigation to a simpler presentation of information related to antibiotic use.
To make sure I answered all of your points in the question, complications that were encountered in these patients were primarily related to COVID-19 (respiratory failure/pulmonary embolism) and a very small fraction (3.2%) of patients developed a hospital-acquired infection, of which the majority were UTIs. No allergic reactions attributed to antibiotics were identified during data collection.
Once again, we really appreciate you taking the time to review our manuscript. We hope our responses are satisfactory.
Sincerely, on behalf of the study team,
Aleksa Despotović

Round 2
Reviewer 1 Report
The manuscript has been improved during the revision process.
Reviewer 2 Report
The manuscript has been improved and is now suitable for publication.